# Study on the releasing regularity of asphalt fume and its suppression technology

**Guang Yang[1], Yongli Xu[1], Xiaolei Jiao[2], Yiming Li[3]***

**1** School of Civil Engineering and Transportation, Northeast Forestry University, Harbin, China, **2** Tianjin Highway Development Service Center, Tianjin, China, **3** School of Civil Engineering, Northeast Forestry University, Harbin, China

* li_yiming2017@163.com

## Abstract

To reduce the unorganized release of asphalt fume during construction, a smoke suppressing and deodorizing (SSD) asphalt was prepared by adding smoke suppressants, which were styrene-butadiene-styrene (SBS), magnesium oxysulfate (MOS) whisker, and deodorant. The effect of smoke suppressants on the release amount of asphalt fume was investigated by mass method, and the effect of deodorant on the odor level of asphalt fume was evaluated by the odor intensity grading method. The results show that when the mass fractions of SBS, MOS and deodorant are 3%, 3% and 0.15%, respectively, 53% of asphalt fume can be effectively reduced, and the odor intensity grade can be significantly lowered. Based on FTIR and SEM analysis, the structural composition of modified asphalt was studied. It shows that SBS can break the dynamic balance of asphalt and produce an uneven network system, and whiskers interspersed in the asphalt play a role in regulating the thermal properties. Besides that, according to the results of thermogravimetry-mass spectrometry (TG-MS), the SSD asphalt can reduce the emission of volatile organic compounds and particulate matter during the high-temperature asphalt production process, with a significant inhibitory effect on small molecules.

## 1 Introduction

With the development of highway construction and the improvement of infrastructure, asphalt pavement has been widely used for various advantages such as comfortable driving, low noise, good skid resistance, and ease of maintenance. However, during the production process of asphalt (mixing, paving, servicing, storage, and transportation), numerous harmful fumes can be released. These particles as air pollutants can directly or indirectly affect the development of the world economy, which has become a matter of widespread concern [1–4].

Asphalt fume contains polycyclic aromatic compounds, among which 4–6 cyclic aromatic compounds are well-known carcinogens. Shicong Mo et al. [5] found 12

**Data availability statement:** All relevant data are within the manuscript and its Supporting information files.

**Funding:** Supported by the Fundamental Research Funds for the Central Universities. (2572022AW55) received by Guang Yang. The funders had no role in study design, data collection and analysis, decision to publish, or preparation of the manuscript

**Competing interests:** The authors have declared that no competing interests exist.

kinds of polycyclic aromatic hydrocarbons (PAHs) pollutants in asphalt fume at 160°C based on CG-MS analysis. The toxicological studies of experts have shown that the fume condensates produced by asphalt will cause carcinogenic skin tumors in mice. The toxicological activity of asphalt fume and fume condensate may exhibit weak initiating activity, which can impact gene expression and immunotoxicity, and has indirect carcinogenic effects [6,7]. In addition to that, the floating particles of asphalt fume in the natural ecosystem may be inhaled into the human lungs, or pollute the soil and adjacent water sources when it rains or snows, and then adversely affect the circulation of the ecosystem, so the pollution of asphalt fume to the ecological environment cannot be ignored [8,9]. However, the existing asphalt fume treatment methods are still concentrated in the combustion method, condensation method, adsorption method, capture method, etc. These methods are mainly applicable to the centralized disposal of fumes in the asphalt production process, but not to the disorganized discharge of asphalt fumes during the large-scale construction of asphalt pavements. Another measure to reduce the release amount of asphalt fume is to add smoke or warm mixing agents to lower the mixing temperature during construction; however, this approach often encounters problems, such as poor odor reduction and a decline in asphalt performance [10,11]. Therefore, whether for human health or environmental protection, it is an imperative trend to develop environmentally friendly asphalt to reduce the fumes and pungent smells produced in the production and construction processes of asphalt pavements [18].

It is a common technique to use flame retardants as asphalt fume suppressants [12]. Xiaolong Yang et al. [13] found that adding ATH and OMMT could effectively inhibit the release of volatile organic compounds in modified asphalt. Some scholars used composites synthesized from expanded graphite and other halogen-free flame retardants, based on the inhibition principle of various flame retardants and the principle of crossing temperature action ranges, to improve the flame retardancy and smoke suppression effects of asphalt [14,15]. Another way is adding polymer, through the formation of a network structure and fixation of internal micro-molecules, thereby strengthening the internal structure of the asphalt binder [16]. Adding adsorbents including physical adsorbents and chemical adsorbents is also an effective way. The porous structure of the adsorbent provides a huge surface area, so there is a large intermolecular force in the pore wall to fix the micro-molecular components [17].

The composition, production mechanism and toxicity of asphalt fume have attracted the attention of scholars, and the enrichment and detection of asphalt fume have become the first problem. Currently, since there is no professional device for asphalt fume enrichment, scholars generally conduct research on asphalt fume through independent enrichment devices [18]. Considering that the amount of asphalt fume in the experiment is small and it is difficult to control, researchers need to use a certain asphalt fume collecting device to analyze it, such as connecting the asphalt fume collecting device to the analysis instrument, or collecting the fume through a filter device, and then testing it through the mass method, CG-MS, TG-MS, liquid chromatography and other methods [19–22].

In this paper, the releasing regularity of asphalt fume and the factors affecting the release of asphalt fume were studied. Based on the synergistic effect of flame retardants and polymer inhibitors, asphalt is modified. Under the premise of ensuring road performance, it can solve the problem of polycyclic aromatic hydrocarbons (PAHs) in the mixing, paving and construction processes of hot mix asphalt to a certain extent, and reduce the release of asphalt fumes and odors during production and construction.

## 2 Materials and methods

### 2.1 Materials

In this study, base asphalt 90# from Liaohe Oilfield, Panjin City, China was used as binder and its properties are shown in the Table 1. The MOS whisker produced by China Kaishefeng Industrial Co., LTD and SBS provided by Dongguan Dingxin Plastic Raw Material Co., LTD were adopted as smoke suppressants, whose properties are shown in Table 2. The deodorant produced by Shanghai Qianbao Fine Chemical Co., Ltd. was used to deal with the pungent odor of asphalt, which is a colorless transparent liquid with slight aroma and its properties are shown in Table 3.

### 2.2 Methods for asphalt fume

**2.2.1 Collection device of asphalt fume.** Based on the mass method and laboratory conditions, an asphalt fume collection device was designed and assembled, which consists of three parts: the asphalt fume generating device, the air supply device and the asphalt fume filtering device. Micron-grade PTFE hydrophobic filter film has an excellent filtration effect on asphalt fume, and it is convenient to install and remove, which can improve the accuracy and convenience of the device. The detailed device composition is illustrated in Figs 1 and 2, of which the three components are shown below.

1. Asphalt fume generating device: The three-mouth flask was used as the asphalt container, the thermostatic heating sleeve was used as the asphalt temperature control device, and the stirrer was equipped with a double-leaf PTFE stirring head to provide the flow conditions for the asphalt and to ensure the overall air tightness of the device.

**Table 1. Properties of base asphalt.**

| Detection index | | Unit | Measured value |
|---|---|---|---|
| Penetration (25 °C, 5 s, 100 g) | | 0.1mm | 82 |
| Softening point (global method) | | °C | 47.3 |
| Ductility (5 °C, 5 cm/min) | | cm | 7.3 |
| Ductility (10 °C, 5 cm/min) | | cm | 30 |
| RTFOT (163°C, 85 min) | Mass change | % | −0.23 |
| | Residual penetration ratio | % | 62.5 |
| | Residual ductility 10 °C | cm | 10.1 |

**Table 2. Properties of MOS whisker.**

| Micromorphology | Length-diameter ratio | Diameter (µm) | Average length (µm) | PH value |
|---|---|---|---|---|
| Acicular fiber | >30 | <1 | 10~60µm | 9.2 |

**Table 3. Properties of deodorant.**

| Appearance | Chemical component | density (g/ml) | Fusing point (°C) |
|---|---|---|---|
| Colorless transparent liquid | Organic compound, polymer of zinc ricinoleate | 1.118 | 18 |

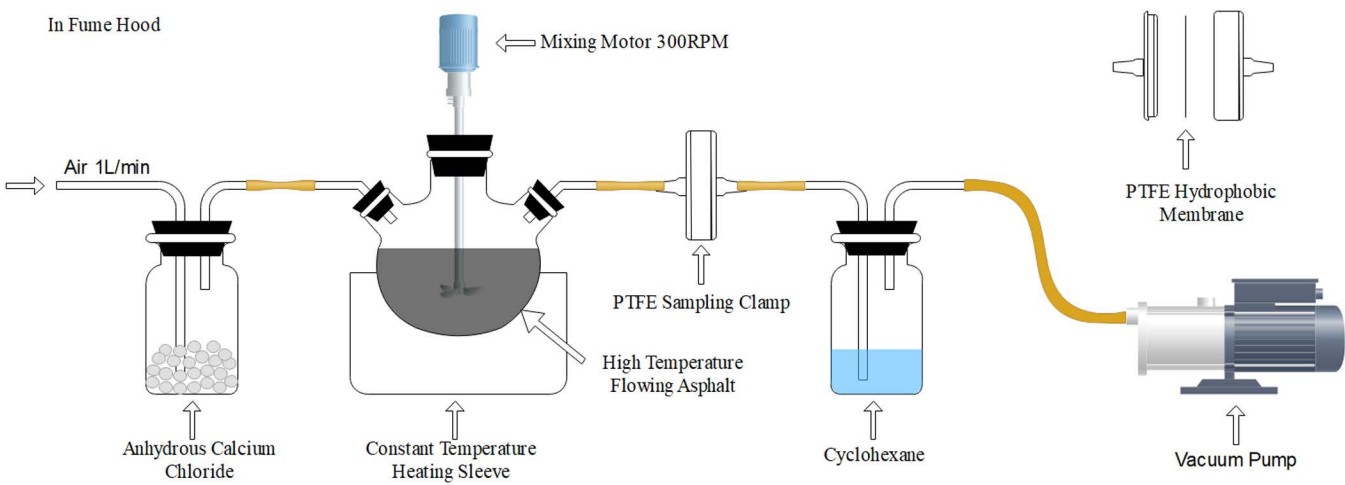

**Fig 1. Asphalt fume collection device.**

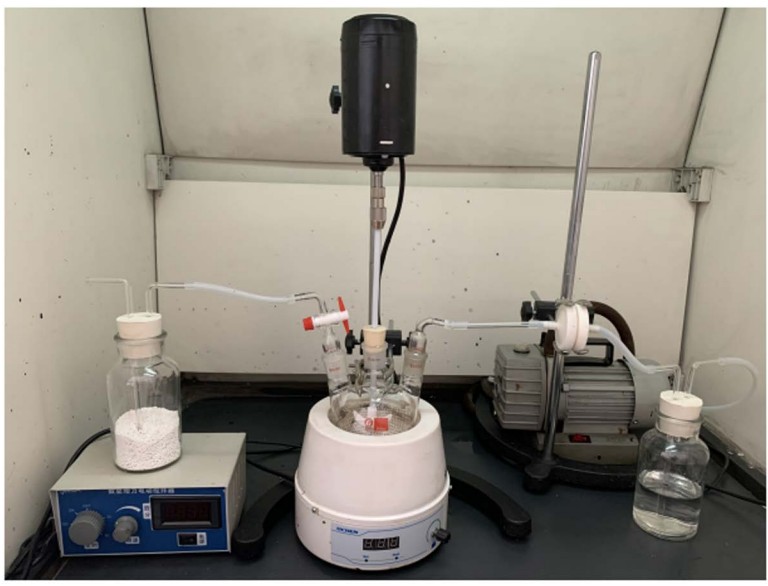

**Fig 2. Picture of the asphalt fume collection device.**

2. Air flow supply device: The sampling pump was equipped with an air flow meter as a constant air flow supply device. The first section of the device was equipped with anhydrous calcium chloride to filter out the moisture in the air to ensure the constant drying of the airflow throughout the whole closed device.

3. Asphalt fume filter device: The microporous PTFE hydrophobic membrane (fixed in the membrane clip) was used to filter volatile organic compounds in asphalt fume under the action of constant air flow.

Compared with the asphalt fume collecting device designed by scholars before, this device improves the asphalt fume adsorption material, upgrading the glass fiber filter cotton, glass fiber filter cylinder, activated carbon and other asphalt

fume adsorption materials in the previous filter device to PTFE hydrophobic filter membrane, and equipping with filter film clip, which effectively improves the adsorption efficiency on asphalt fume.

The filter material used in the study was a PTFE hydrophobic filter membrane with a pore size of 0.1μm. Compared with traditional glass fiber filter cotton, it has the advantages of high voids, low air resistance, and high flux. At the same time, the suction filtration effect can be increased by 6–7 times. Even under a very low-pressure difference, it can ensure the unimpeded passage of air or other gases. Besides that, it has excellent chemical compatibility, a high retention rate and high temperature resistance and it is simple to operate. In the whole test process, the PTFE hydrophobic filter membrane was placed in the openable filter membrane clamp as shown in Fig 3. As an important part of the complete filter device, the filter membrane absorbed and filtered the asphalt fume under the action of the air supply device. The surface of the filter membrane after suction filtration is yellow and slightly wrinkled, as shown in Fig 4. The left half of it is the

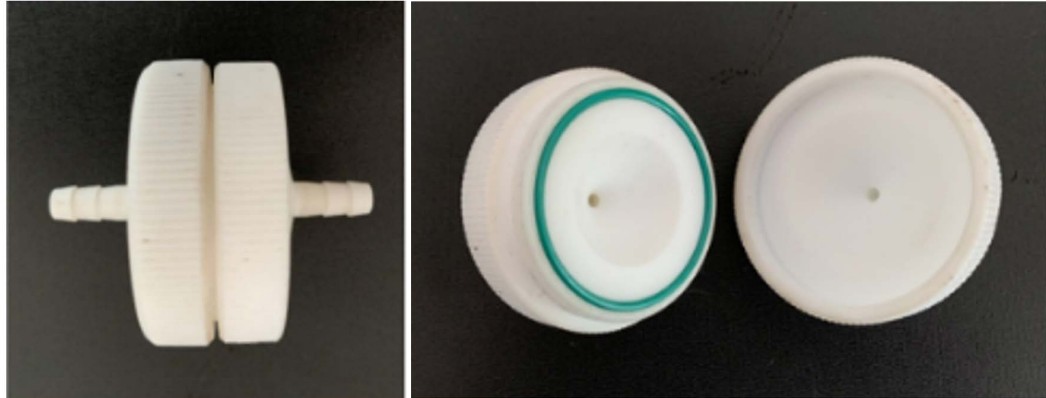

(a) Filter membrane clamp closed          (b) Filter membrane clamp opened

**Fig 3. Filter membrane clamp.**

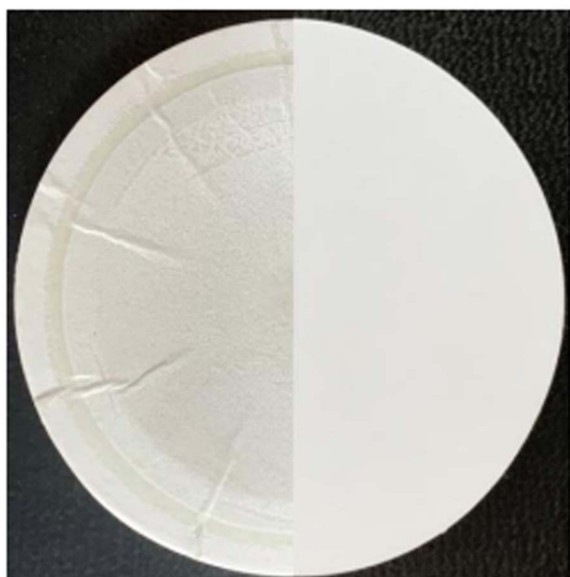

**Fig 4. Filter membrane.**

filter membrane for adsorbing fume, and the right half is the clean filter membrane. The yellowish-brown substance is the filtered fume.

**2.2.2 Influencing factors of asphalt fume release.** Since the influence factors of the release amount of asphalt fume under high temperatures are complex, the influence of the type of asphalt, heating time, heating temperature and mixing speed was studied. To simulate the construction temperature, the temperature of the asphalt was controlled within the range of 120–180°C during the whole process, and the air flow rate was maintained at 1L/min. The influence of asphalt composition was ignored since only one kind of asphalt was adopted. Under different experimental conditions as shown in Table 4, asphalt fume was collected using the previously described asphalt smoke collection device to explore the effects of temperature, heating time, and stirring speed on the asphalt fume generation.

**2.2.3 Determination of the release amount of asphalt fume.** To ensure the accuracy of the test data, all glassware was washed with cyclohexane before the test and placed in a vacuum drying box with PTFE hydrophobic filter membranes at 80°C for 3h.

200g asphalt was injected into the three-port flask. Then the three-port flask containing asphalt was put into the heating sleeve, and the temperature sensor and the agitator were inserted and fixed. In the process of inserting the agitator, the mixer head with Teflon was opened with the help of the inner walls to ensure that the mixing head is completely immersed in the asphalt. After that, the speed of the agitator and the temperature of the heating sleeve were adjusted until the temperature was constant with the required test temperature, which is generally within the range of 120–180°C.

The dry PTFE hydrophobic filter membrane was weighed and its net weight was recorded. And then the filter membrane was placed in the filter membrane clip and tightened, whose smooth surface was the filter surface, and the air flow entered through the smooth surface and exited through the rough surface during the test.

When the temperature was stable, timing was started at the moment when the agitator and vacuum pump were opened, and the PTFE hydrophobic filter membrane was replaced with a new one within the optimal filtration time. When the filter membrane was being replaced, the vacuum pump was switched off and the timing was paused. After the replacement was complete, the timing continued and the filter device was started simultaneously. After the required test time was reached, the filter membrane filled with asphalt fume was weighed. The mass change of the filter membrane before and after filtration was determined as the mass of asphalt fume. The whole test lasted 1-3h.

Finally, the asphalt fume mass was calculated according to formula 1, in which $m_1$ is the total mass of the filter membrane with the asphalt fume within the specified time, $m_2$ is he total mass of the dry filter membrane before the filtration within the specified time, and $\triangle m$ is the total mass of the asphalt fume adsorbed within the unit time.

$$\Delta m = m_1 - m_2$$

(1)

**2.2.4 TG test.** In this study, TG test was conducted using a STA449F3 thermal analyzer and a QMS403D mass spectrometer to evaluate and compare the volatile hazardous substances in pure flavor smoke suppression asphalt and matrix asphalt. During the test, 10 mg of each asphalt sample was put into a crucible, which was then inserted into a comprehensive thermal analyzer. It was heated to 200°C at the rate of 10°C/min, with nitrogen as the protective gas. Three samples of each asphalt were tested, and the average ionic current intensity at the same temperature was calculated and reported. To reduce error, all 10 mg samples were sampled from the asphalt surface. To guarantee the

**Table 4. Condition parameters.**

| Test plan | Heating temperature (°C) | Heating time (h) | Mixing speed(r/min) |
|---|---|---|---|
| 1—Varying heating temperature | 120,140,160,180 | 2 | 300 |
| 2—Varying heating time | 160 | 1,2,3,4,5 | 300 |
| 3—Varying mixing speed | 160 | 2 | 0,100,200,300,400 |

accuracy of smoke testing, the base asphalt utilized in the preparation of modified asphalt shares the same heating history.

**2.2.5 DSC test.** The heat flow type differential scanning calorimeter produced by TA Company of the United States was adopted as the test instrument. The heating rate was 10°C/min, the highest test temperature was 19°C, and nitrogen was used as the protection gas. Since DSC is a high-precision instrument, the sample requirements are stringent. The mass of the sample should not exceed 5 mg, and the weighing should be of high precision. The sample was weighed using a high-precision balance with an accuracy of 0.001. The specimen shape is circular. The base asphalt utilized in the preparation of modified asphalt shares the same heating history.

## 2.3 Preparation and performance analysis of modified asphalt

**2.3.1 Preparation.** The asphalt was heated to 160°C in a constant-temperature oven and a high-speed shear machine was used to prepare the modified asphalt. According to the asphalt quality and the pre-designed mixing ratio, the quality of SBS, MOS, and deodorant was weighed. The modification was divided into three steps: Firstly, SBS was added to the asphalt at 160°C and evenly stirred for 15 min. After that, the asphalt was put into the shear machine at 160°C and 6000 r/min to be sheared for 30 min. Secondly, the asphalt was taken out, and MOS was added at 150°C and stirred evenly for 15 min. Then they were sheared for 30 min at 150°C and 4000 r/min. Finally, the deodorant was dropped and stirred at 150°C for 20 min.

**2.3.2 Performance analysis.** In this study, the properties of modified asphalt were determined in accordance with the Chinese standard JTG-E20-2011. There were three samples for each trial and the average was taken. The road performance of the asphalt mixture was also evaluated, including high temperature stability (dynamic stability), low temperature crack resistance (maximum bending tensile strain) and water stability (residual stability and residual strength ratio).

**2.3.3 Odor intensity grading.** The asphalt odor was evaluated based on the Japanese odor intensity grading method. Twenty volunteers were invited to score the asphalt odor according to the odor intensity grading standard, and the average value was finally taken. The asphalt was heated to 160°C and the fan smell method was used to measure the odor intensity within the safe distance. Each group of data was measured three times, and finally the data were summarized. The base asphalt utilized in the preparation of modified asphalt share the same heating history.

**2.3.4 Infrared spectrum analysis.** The asphalt samples were analyzed by a Fourier Transform Infrared Spectrometer (Spectrum 400). The sample was prepared by the KBr compression method. In the transmittance mode, there were 120 scans per spectrum, with a scanning wavelength range of 500 cm⁻¹–4000 cm⁻¹.

**2.3.5 Scanning electron microscope (SEM).** Asphalt samples were paved on the loading plate, and the surface layer of the asphalt was sprayed with gold using the vacuum coating method. Then the asphalt samples were taken under the lens of a scanning electron microscope. The compatibility of modifiers, surface morphology, fracture morphology and fiber structure of different asphalt samples was characterized by three-dimensional morphology scanning. The type of scanning electron microscope used is JSM-7500F, and the magnification is 1000 times.

**2.3.6 Thermogravimetric mass spectrometry (TG-MS).** Thermogravimetric mass spectrometry analyzer (TG-MS) consists of a thermogravimetric analyzer and a mass spectrometer, which can monitor the weight loss and original emission data of samples during the pyrolysis and combustion process in real time, and output TG/DSC data. In this experiment, NETZSCHSSTA449F3 (-QMS403D-IS50) TG/DSC synchronous thermal analyzer was combined with a mass spectrometer. A 10 mg asphalt sample was placed at the bottom of the crucible. The test temperature range was set as 35–200°C and the heating rate was 10°C/min. Nitrogen was used as the inert protective atmosphere inside the thermogravimetric analyzer, and the flow rate was 0.1L/min. After being tested by the thermogravimetric analyzer, the reaction products were introduced into the mass spectrometer for further analysis. The ionic current intensity at each temperature was calculated and reported for the scanned material. TG-MS analysis cannot accurately obtain the inhibition

amount of each substance, but the inhibition effect on asphalt fume can be characterized and evaluated based on the ionic current intensity at each temperature. The base asphalt utilized in the preparation of modified asphalt share the same heating history.

**2.3.7 The process diagram.** Due to the involvement of numerous experiments, for ease of understanding, the research ideas and technical route of this paper are shown in Fig 5.

## 3 Results and discussion

### 3.1 Influencing factors of asphalt fume release

Results show that temperature, stirring speed and heating time are the main influencing factors, among which heating temperature is the most sensitive factor. As shown in Fig 6, the release amount and its growth rate of asphalt fume gradually increase with the increase of temperature, and 140°C is a dividing point. Before 140°C, the release amount of asphalt fume is small and the growth rate is slow, while after 140°C, the release amount of asphalt fume is large and the growth rate is fast. The influence of heating time is shown in Fig 7. The release amount of asphalt fume shows the characteristic of gradual attenuation, which indicates that the release rate of asphalt fume increases with the increase of time and gradually tends to be stable. The influence of the mixing speed can be seen in Fig 8. The release amount of asphalt fume increases with the increase of the mixing speed and gradually reaches the upper limit of the release amount, where the increase of the mixing speed has little effect on the release of asphalt fume.

Considering the actual processing conditions in asphalt pavement engineering, the release law of asphalt fume was summarized as the test conditions that the heating time was 3h, the heating temperature was 160°C, and the mixing speed was controlled at 300r/min, which was adopted in the subsequent asphalt modification.

### 3.2 The influence of smoke suppressants

**3.2.1 The influence of MOS.** MOS is produced by hydrothermal synthesis of magnesium oxide and magnesium hydroxide evenly dispersed in magnesium sulfate solution under high temperature and pressure. As an inorganic flame retardant and smoke suppressant, MOS has been widely used in plastics, paper and wood, etc. Besides that, with a

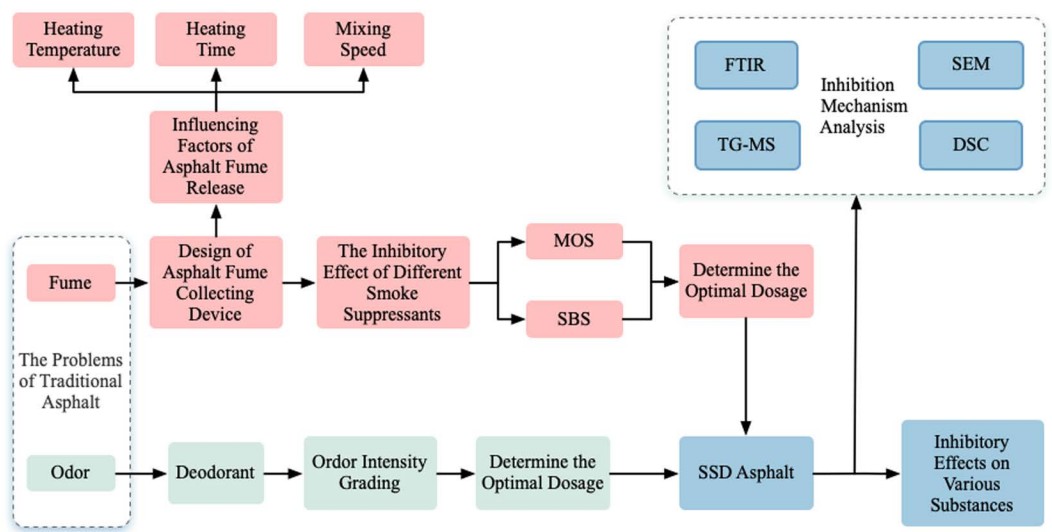

**Fig 5. The process diagram.**

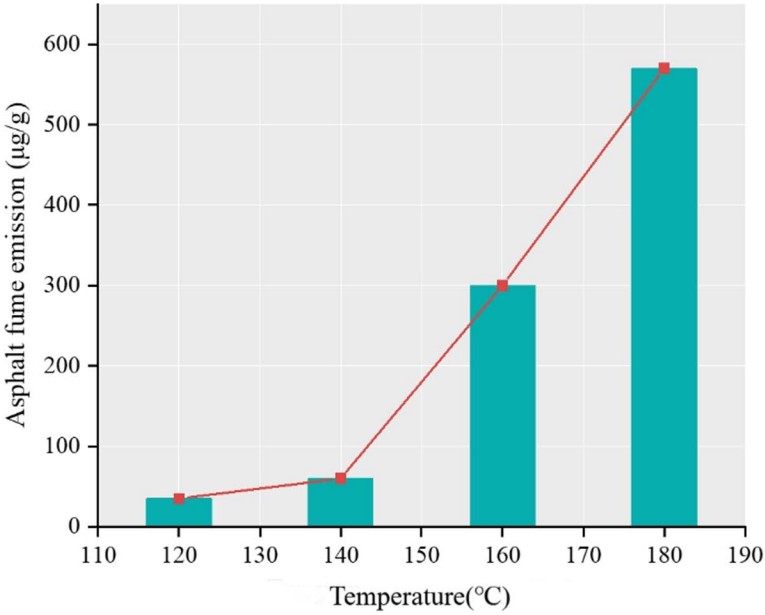

**Fig 6. Asphalt fume emission at different heating temperatures.**

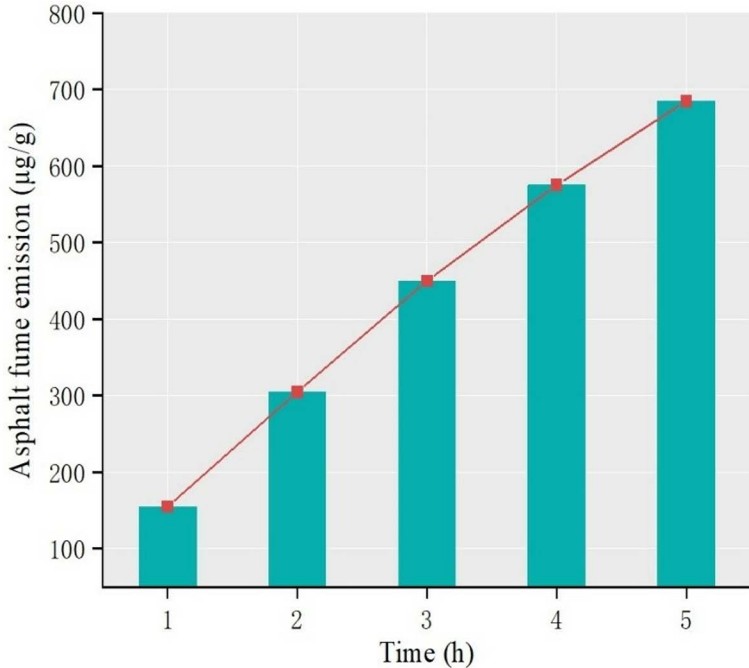

**Fig 7. Asphalt fume emission at different heating time.**

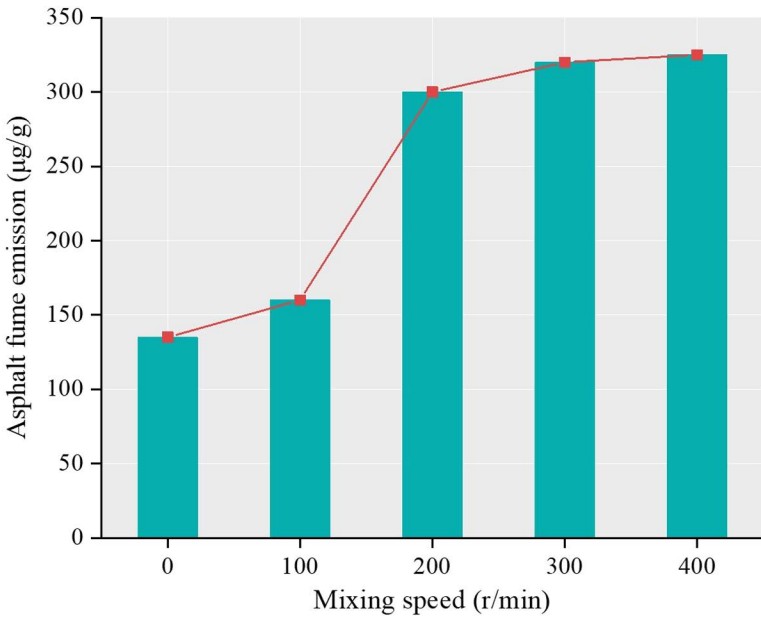

**Fig 8. Asphalt fume emission at different mixing speeds.**

certain breaking strength and elastic modulus, MOS is also an excellent reinforcing agent. Since the whisker size is micro/nano level, whose diameter is about 0.8–0.12μm and the length is about 20–200μm, it has good compatibility with asphalt so that it can be evenly distributed in asphalt. However, excessive MOS will decrease its dispersibility in asphalt, which will lead to a decrease in the ductility of asphalt. In this study, on the premise of ensuring performance of asphalt, the six contents of MOS (0.5%, 1%, 1.5%, 2%, 2.5%, and 3%) were selected. The inhibition rate (IR) was calculated according to formula 2, in which $\Delta m_{NA}$ is the total mass of fume from base asphalt and $\Delta m_{MA}$ is the total mass of fume from asphalt modified with MOS or SBS. The results are shown in Fig 9.

$$R = \frac{\Delta m_{NA} - \Delta m_{MA}}{\Delta m_{NA}}$$

(2)

According to the data in Fig 9, the inhibition rate of asphalt fume increases with the increase of MOS content and the relationship is close to linear. However, when the content of MOS is greater than 3%, it affects the penetration and ductility of asphalt. The smoke suppression mechanism of MOS involves releasing bound water and absorbing a lot of heat energy when heated and decomposed, so as to absorb a lot of latent heat in the asphalt. Compared with magnesium hydroxide and magnesium oxide, the combined water volume of MOS is larger, and the generated water vapor can not only affect the heat absorption and release of asphalt, but also reduce the smoke concentration on the surface of asphalt, so as to achieve the effect of smoke elimination.

**3.2.2 The influence of SBS.** SBS is a common asphalt modifier of asphalt, which can effectively improve the high and low temperature properties, tensile properties, and elasticity of asphalt [23,24]. In this study, on the premise of ensuring the performance of asphalt, the eight contents of SBS (0.5%, 1%, 1.5%, 2%, 2.5%, 3%, 3.5%, 4%) were selected. When SBS shear is mixed with asphalt, due to the incompatibility of polystyrene and polybutadiene at room temperature, the cohesion-energy density of polystyrene between molecular chains in the copolymer is high. Therefore, the two ends of the polystyrene are first gathered together with other polystyrene, forming a physical cross-linking region of many constraining

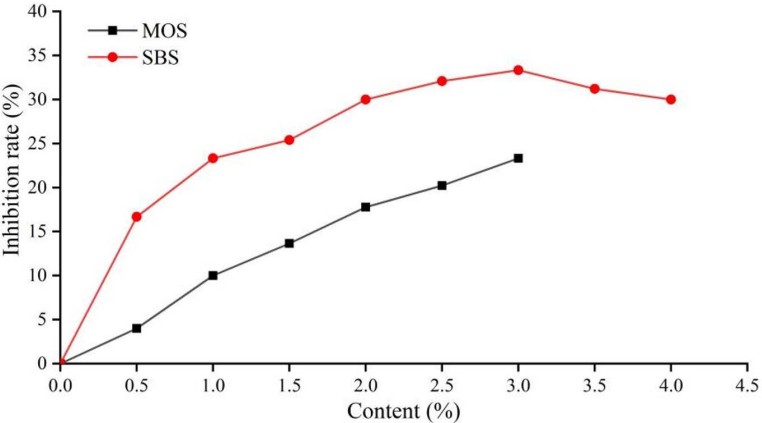

**Fig 9. Inhibition rate of MOS and SBS on asphalt fume.**

components. Since the blocks are flexible polybutadiene, a network structure is formed. When the SBS exists in the form of a physical cross-linking network structure in the asphalt, the SBS particles affected by saturates and aromatics in the asphalt swell and evenly dispersed in the asphalt. As a result, the proportion of asphalt components changes and the dynamic balance of asphalt is broken, resulting in an uneven system, thus the volatilization of active components of asphalt is effectively inhibited. The composition ratio of asphalt is changed and the dynamic balance of the original asphalt is broken, resulting in an uneven system, which can effectively inhibit the volatilization of the active components of asphalt and have a significant effect on the inhibition of asphalt fume.

**3.2.3 Synergistic effect of SBS and MOS.** Based on the above data, the content range of MOS was 0–3%, and the content range of SBS was 0–4%, both with the steps of 1%. The experiment was designed as shown in Table 5. To investigate the influence of their content on asphalt fume, the three indexes of asphalt experiment and asphalt fume collection experiments were carried out, and the inhibition rate-content 3D projection map was drawn (Fig 10). The results show that when the two smoke suppressants are combined, the inhibition rate of asphalt fume is greater than that of the single modification. The inhibitory effect can be illustrated more intuitively by the photo of the filter membrane as shown in Fig 11. The four parts are all direct shot images of the filter membranes used for four typical types of asphalt with a filtering time of 15 min. The upper right is base asphalt, the lower right is MOS modified asphalt, the upper left is SBS modified asphalt, and the lower left is asphalt modified by MOS and SBS. Based on the shadow area and color depth of asphalt fume, it can be seen that the inhibitory effect on asphalt fume is as follows: MOS and SBS > SBS > MOS > base asphalt. Among which the fume of asphalt modified by MOS and SBS has only a slight yellow shadow on the filter membrane, which indicates that their compound modification has a much better suppression effect on asphalt fume. It is speculated that SBS and MOS can constitute a cooperative fume suppression system, which effectively improves the inhibitory effect on asphalt smoke.

It is worth mentioning that, similar to previous studies, when SBS and MOS are mixed, the optimal dosage of SBS is 3%. If the dosage of SBS is 4% or more, the penetration of asphalt will be poor and not meet the performance requirements. When the dosage of SBS is 3%, a dense network structure has been formed, and the addition of more SBS would be easy to widen the polymer interstice, increase the softening point, and reduce the penetration and ductility. When MOS whisker and asphalt are used in combination, in addition to playing a role as flame retardant, the fine structure of a whisker makes it evenly distributed in the mesh structure of asphalt, making the structure more compact. The asphalt and swelled SBS have a bone-like whisker inserted inside, which makes them more stable when heated. The greater specific surface area of the whisker also greatly improves the heat absorption effect. Based on the MOS and SBS synergistic

**Table 5. The performance of MOS and SBS compound modified asphalt.**

| MOS (%) | SBS (%) | Penetration (0.1 mm) | Softening Point (°C) | Ductility (cm) | Asphalt fume emission-3h (ug/g) | Inhibition rate (%) |
|---|---|---|---|---|---|---|
| 0 | 0 | 82 | 46.3 | 7.3 | 450 | 0.00% |
| | 1 | 74.5 | 52.2 | 10.5 | 345 | 23.33% |
| | 2 | 69.5 | 52.4 | 16.5 | 315 | 30.00% |
| | 3 | 64.5 | 53.2 | 23.5 | 300 | 33.33% |
| | 4 | 60 | 55.1 | 30 | 315 | 30.00% |
| 1 | 0 | 76.3 | 48.2 | 5.6 | 405 | 10.00% |
| | 1 | 74 | 49.5 | 13.6 | 270 | 40.00% |
| | 2 | 63.3 | 52 | 12.4 | 285 | 36.67% |
| | 3 | 62.6 | 53.4 | 11.6 | 270 | 40.00% |
| | 4 | 57.8 | 54.4 | 19.1 | 270 | 40.00% |
| 2 | 0 | 76.3 | 48.9 | 5 | 370 | 17.78% |
| | 1 | 72.5 | 54.2 | 10.4 | 255 | 43.33% |
| | 2 | 66.9 | 51.1 | 10.7 | 240 | 46.67% |
| | 3 | 64.9 | 53.4 | 15.8 | 225 | 50.00% |
| | 4 | 58.8 | 50.3 | 12.5 | 240 | 46.67% |
| 3 | 0 | 75.9 | 49.1 | 4.3 | 345 | 23.33% |
| | 1 | 71 | 50 | 9.8 | 255 | 43.33% |
| | 2 | 64.4 | 53.5 | 11.7 | 240 | 46.67% |
| | 3 | 62.7 | 55.5 | 12.4 | 210 | 53.33% |
| | 4 | 54.8 | 56.5 | 11.9 | 225 | 50.00% |

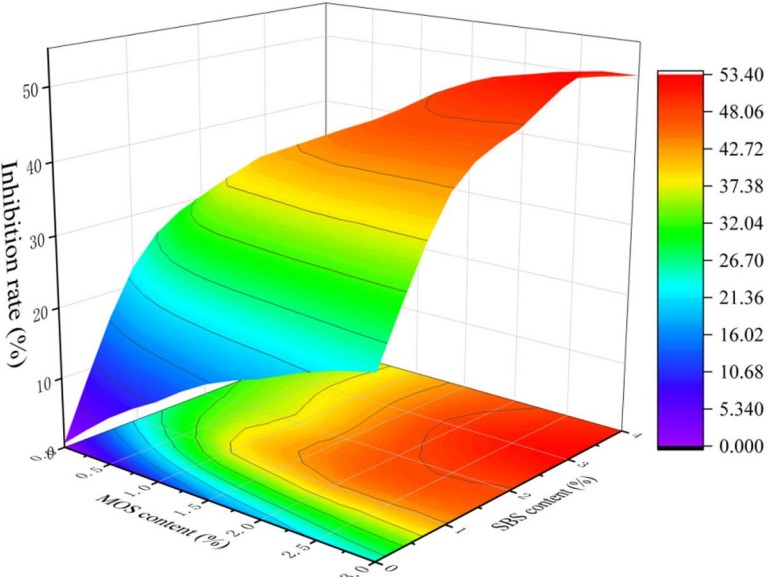

**Fig 10. Synergistic effect of SBS and MOS on inhibition rate.**

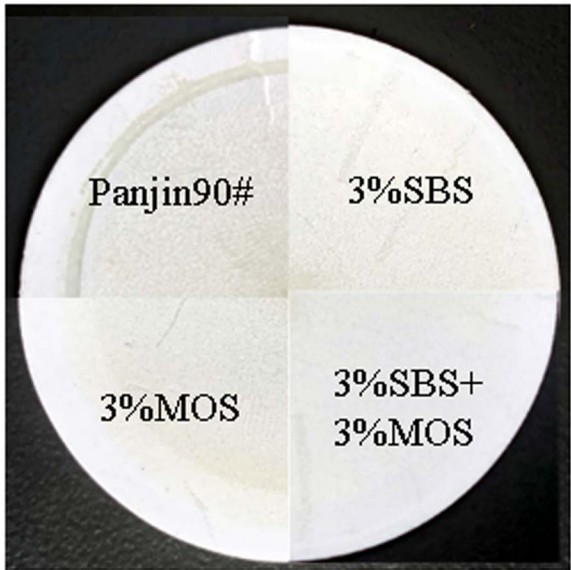

**Fig 11. Photo of filter membranes used for four typical types of asphalt.**

smoke suppression system and the experimental results, 3% MOS and 3% SBS were found to be the optimal dosages, with a suppression rate of 53%.

### 3.3 The effect of deodorant

Asphalt with deodorant contents of 0‰, 1‰, 1.5‰ and 2‰ was evaluated using the Japanese odor intensity grading method, and the results are shown in Fig 12.

　The data show that under the high temperature condition of 160°C, 1‰ deodorant can reduce the odor level of base asphalt from 4- strong odor to 1- barely sensed odor, and its fragrance can play a certain role in covering. When the

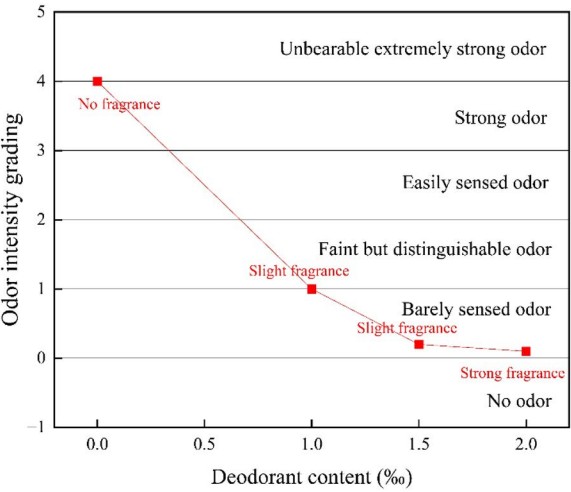

**Fig 12. Effect of deodorant.**

dosage reaches 1.5‰, the odor level of base asphalt decreases to 0.2, which is close to odorless. If the dosage of deodorant is increased to 2‰, the odor intensity reaches 0.1. It can be seen that the odor intensity is not significantly improved when the dosage increases from 1.5‰ to 2.0‰. In addition, according to the design principle of SSD, excessive fragrance may have an adverse effect on the operator. Based on the above reasons, the optimal dosage of deodorant was determined as 1.5‰.

Based on the above research, the final composition of the modifier is determined as follows: 3% MOS, 3% SBS and 1.5‰ deodorant. The modified asphalt with this combination was named smoke suppressing and deodorizing (SSD) asphalt.

### 3.4 Road performance of SSD asphalt

**3.4.1. Asphalt performance analysis.** According to the test procedures, the three indexes of base asphalt, SBS modified asphalt, MOS modified asphalt, deodorant modified asphalt and SSD asphalt were tested, and each group of tests was repeated three times. The results are shown in Fig 13.

The test data show that the SSD asphalt meets the requirements of the specification and has good properties, whose softening point elongation is stronger than base asphalt, and the needle penetration is slightly lower than that of the base asphalt. The penetration data indicate that the addition of MOS whiskers or SBS can affect the consistency of the asphalt, making it more viscous. The low temperature properties of the five asphalts are ranked as follows: SBS modified asphalt> SSD asphalt＞base asphalt> deodorant modified asphalt>MOS modified asphalt, which indicats that the positive effect of SBS on the ductility was stronger than the negative effect of basic magnesium sulfate whisker, and their combined modification can improve the low temperature performance of asphalt. In terms of softening point, the softening point of SSD asphalt is higher than that of the other four kinds of asphalt, which indicates that SSD suppressant asphalt has the highest high temperature stability. In addition, compared with deodorant modified asphalt, base asphalt has little difference in the three indexes, which indicates that deodorant has little effect on the conventional performance of asphalt.

**3.4.2 Analysis of road performance of asphalt mixture.** The dynamic stability of the five mixtures in Fig 14a all meet the technical indicators required by the specification, and the order of dynamic stability is as follows: SSD asphalt> SBS modified asphalt> MOS modified asphalt> base asphalt> deodorant modified asphalt. Among them, the dynamic stability of the SSD asphalt mixture reaches more than 3 times that of the base asphalt mixture, and the value was 4502.3 times/mm. The MOS whisker can slightly increase the dynamic stability of the asphalt mixture by 15%. Besides that, SBS modified asphalt has obvious improvement in high temperature performance, and the dynamic stability can be increased by more than 2 times. The deodorant would slightly reduce the high temperature stability of the asphalt mixture by only 2%. The experiment shows that the combined application of SBS and MOS whiskers can significantly enhance the heat resistance of SSD asphalt mixture under high temperature conditions, which is also supported by the DSR data.

As can be seen from Fig 14b, the residual strength ratios of the five asphalt mixtures all meet the requirements of the specifications. It can be seen that MOS whiskers and SBS can both improve the water stability performance of the asphalt mixture, while deodorant can reduce the water temperature qualitative energy of the asphalt mixture. When these three modifiers are used for compound modification, the water stability of the SSD asphalt mixture is generally improved, and the water stability performance is greater than that of the other four groups of mixtures. In general, SSD asphalt mixture has excellent water stability and can be used in actual pavement engineering.

It can be seen from Fig 14c that the maximum bending tensile strains of the five asphalt mixtures all meet the specification requirements. The low-temperature performance of the SSD asphalt is the highest, and the maximum flexural and tensile strain is 1.4 times that of the base asphalt mixture, and slightly greater than that of SBS modified asphalt mixture. It can be seen that the SBS modification significantly improves the low temperature performance of the mixture, and the MOS whisker slightly improves the low temperature performance of the asphalt mixture, while the deodorant has no

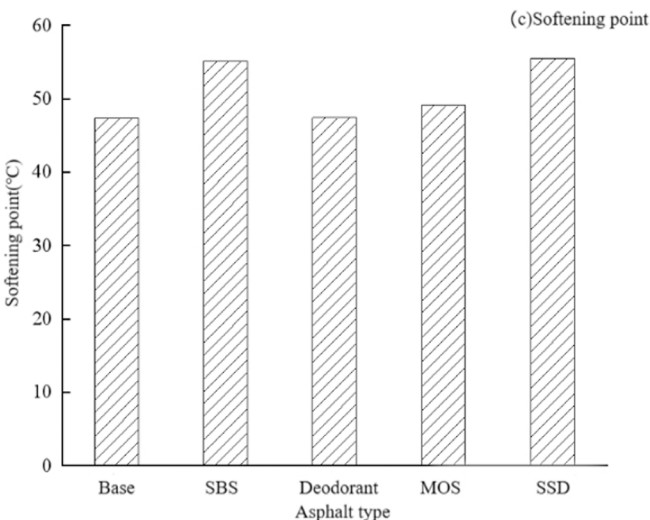

**Fig 13. Asphalt performance.**

obvious effect on the low temperature performance of the mixture. In general, the SSD asphalt mixture has the ability to serve in cold areas and resist low temperature cracking.

### 3.5 Infrared spectrum analysis

As shown in Fig 15, the five groups of infrared spectrum curves show that the base asphalt has obvious absorption peaks at 2924 $cm^{-1}$, 2853 $cm^{-1}$, 1602 $cm^{-1}$, 1459 $cm^{-1}$, 1376 $cm^{-1}$, 1263 $cm^{-1}$, 1032 $cm^{-1}$, 869 $cm^{-1}$, 813 $cm^{-1}$, 741 $cm^{-1}$. The wide absorption peak between 2924 $cm^{-1}$ and 2853 $cm^{-1}$ is caused by the symmetric and antisymmetric stretching vibrations of alkane methylene $CH_2$ in asphalt. The peak at 1602 $cm^{-1}$ is the stretching vibration absorption peak of the toluene $C=C$

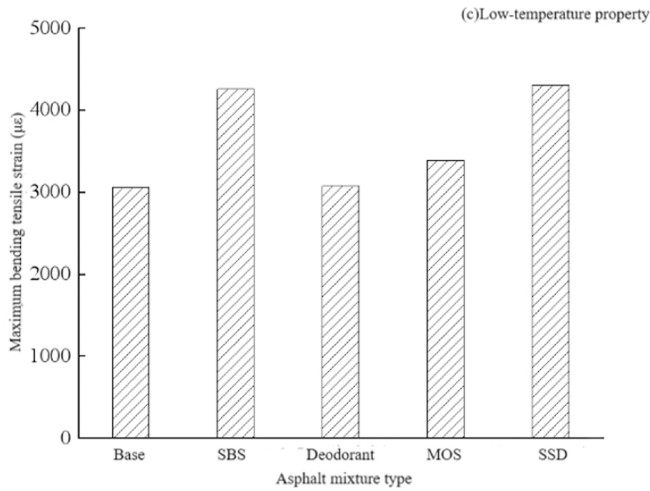

**Fig 14. Asphalt mixture performance.**

double bond in asphalt. The peaks at 1459 cm⁻¹, 1376 cm⁻¹ and 1263 cm⁻¹ are caused by the deformation vibration absorption peaks of -CH$_2$ and -CH$_3$, respectively. The absorption peaks at 869 cm⁻¹, 813 cm⁻¹ and 741 cm⁻¹ were caused by the rocking vibration of C-H on the benzene ring, and the above characteristic absorption peaks were all caused by the typical asphalt structure. It can be seen from the curve in Fig 15 that the infrared spectrum of the modified asphalt is the direct superposition of the infrared spectrum results of MOS, SBS and deodorant.

In the infrared spectrum of MOS and SBS, it can be seen that the characteristic absorption peak of $SO_4^{2-}$ in the MOS is generated at 1117 cm⁻¹, 639 cm⁻¹ and 598 cm⁻¹ respectively, and the polybutadiene CH＝CH is generated at 966 cm⁻¹, which is the characteristic absorption peak of SBS modified asphalt. The characteristic absorption peak of polybutadiene CH＝CH was generated at 966 cm⁻¹, which was the characteristic absorption peak of SBS modified asphalt, and the absorption peaks of 1117 cm⁻¹, 966 cm⁻¹, 639 cm⁻¹ and 598 cm⁻¹ did not exist in the infrared spectrum of asphalt, all of

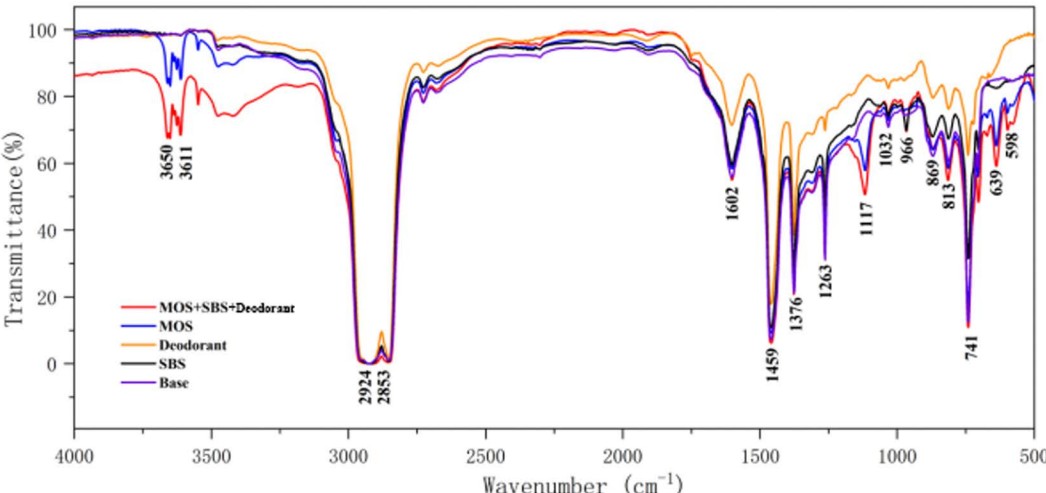

**Fig 15. Infrared spectra of five types of modified asphalt.**

which were new absorption peaks. The spectrum of deodorant is basically similar to that of base asphalt, which indicates that deodorant absorbs odorous substances through physical action. Besides that, the MOS whisker, SBS and deodorant all have an effect on the structure of asphalt, among which the MOS whisker and SBS are combined with asphalt in the form of chemical adsorption.

### 3.6 SEM analysis

As shown in Fig 16, the SEM morphology observation figure suggests that both kinds of asphalt have good homogeneity. The surface of the base asphalt is smooth and has no special morphological characteristics. The surface of the modified asphalt has a wrinkled network structure, which means that the SBS modification makes the SSD asphalt have a larger specific surface area, which in turn enhances the adhesion between asphalt molecules, thereby inhibiting the volatilization of small molecules. Besides that, only a few MOS whiskers are exposed on the surface of the asphalt even under the

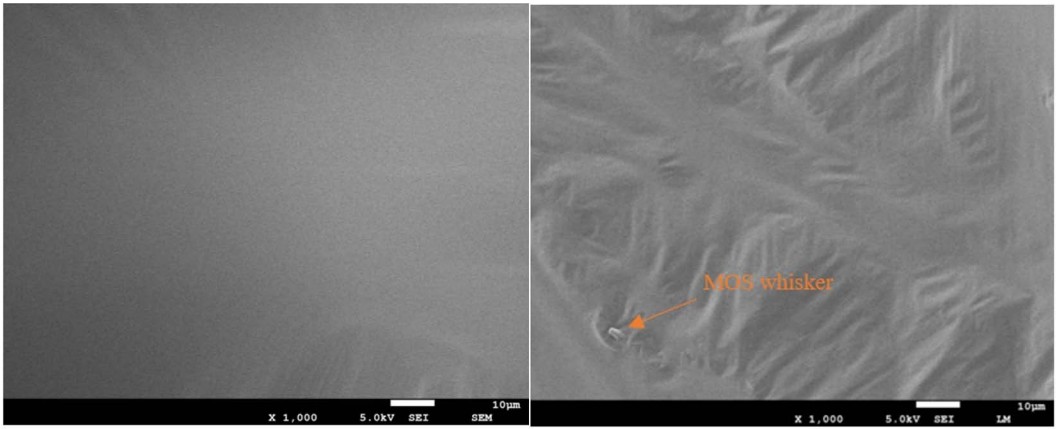

**Fig 16. The surface morphology of base asphalt and SSD asphalt.**

scanning condition of 1000 times, which indicates that the MOS whiskers have very high compatibility with asphalt. It can be concluded that the whiskers are inside the asphalt and they can be inserted into the special structure produced by SBS absorbing aromatics, so they can play a flame-retardant role to reduce the release of asphalt fumes.

### 3.7 TG analysis

The release of volatile organic compounds from asphalt is closely related to the thermal stability of asphalt. The thermal stability characteristics of asphalt materials were studied based on TGA. It can be seen from Fig 17 that the thermogravimetric analysis diagrams of the five types of asphalt are similar. The whole weight loss process can be roughly divided into three stages. The first stage is at 0–300 °C, and the weight loss rate is about 1%. The second stage is at 300–500 °C, in which the weight loss is faster and the weight loss rate is about 82%. In the third stage that is above 500 °C, the weight loss rate tends to be stable, and the weight rate is about 1%. During the whole weight loss process, the thermal weight of the SSD asphalt remained the highest. The thermogravimetric curves of base asphalt and the deodorant modified asphalt are the lowest and similar, which indicates that the deodorant only produces an effect of deodorant on the thermal properties of asphalt, while the addition of SBS and MOS increases the thermal severity of the asphalt. The residual mass of SSD asphalt is 16.5%, and that of base asphalt is 15%, which proves that the SSD asphalt has certain thermal stability, which is an important factor in reducing the volatilization of light components of asphalt, and reducing its flame retardant properties compared to base asphalt.

### 3.8 DSC analysis

Since asphalt is a thermally sensitive material. Differential scanning calorimetry (DSC) was used to study the heat flow characteristics during the pyrolysis of base asphalt and SSD asphalt. As shown in Fig 18, before the beginning of pyrolysis, the base asphalt exhibits heat absorption in the range of 42–83°C, and SSD exhibits heat absorption in the range of 39–118°C. When the temperature exceeds the upper limit of these two ranges, the asphalt begins to release heat. Compared with the base asphalt, the heat flow of the SSD asphalt is significantly reduced in the combustion process based on the modification of smoke suppressants and deodorants. Less heat flow leads to fewer volatiles released from the asphalt in the pyrolysis process. Meanwhile, the SSD asphalt has excellent heat insulation performance. In order to quantitatively analyze the heat flow characteristics of the SSD asphalt, the two curves were integrated to calculate the heat absorption

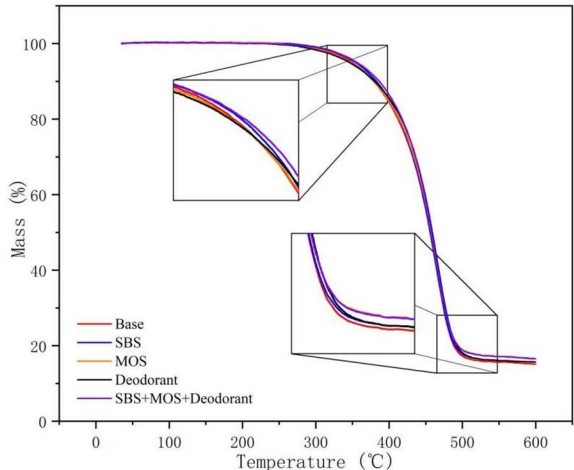

**Fig 17. TGA of five types of asphalt.**

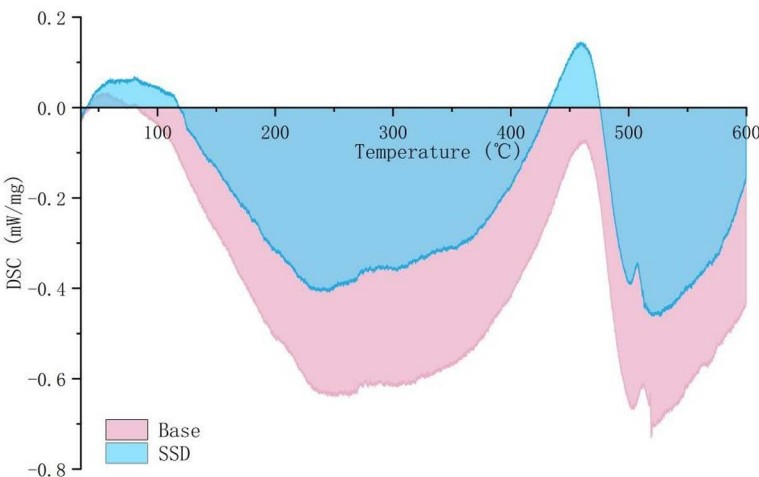

**Fig 18. DSC curve of SSD asphalt and base asphalt.**

and heat release of the two bitumens (the integral above the X-axis is heat absorption, and the integral below is heat release). The heat absorption of the SSD asphalt is 7.28J/g and the heat release is 125.01 J/g. The heat absorption of the base asphalt is 0.63 J/g and the heat release is −229.50 J/g. Compared with the control group, the heat absorption of the SSD asphalt is increased by 1055.6% and the heat release is decreased by 45.5%. All these indicate that the SSD asphalt has higher thermal stability and lower temperature sensitivity, which is also an important reason for the reduction of asphalt smoke release at high temperatures. The results show that the addition of smoke suppressant can reduce the heat release of asphalt combustion system to prevent further combustion of asphalt materials and stabilize small molecules.

### 3.9 TG-MS analysis

Based on previous studies, harmful substances in asphalt fume can generally be divided into three types: small molecules (water, carbon dioxide, etc.), polycyclic aromatic hydrocarbons (potentially harmful substances in asphalt smoke) and other organic substances (chlorobenzene, etc.) [19,25–28]. We took the typical substances in each type and TG-MS was used to carry out specific scanning of these substances based on mass to charge ratio, and the scanning temperature range was 35–200°C. The inhibitory effect of different inhibitors on the release of volatile organic compounds in asphalt was studied,

Fig 19 shows the small molecular substances released from asphalt, and the mass to charge ratio from left to right is in order of $H_2O$, $H_2S$, $CO_2$, $NO_2$, $SO_2$:18, 34, 44, 46, 64, in which ionic current intensity of m/e = 18 (water), m/e = 44 (carbon dioxide) is high because of the moisture of asphalt, crystal water of MOS lose due to heat, and the decomposition of asphalt organic compounds under high temperature conditions. Due to the elemental composition of asphalt itself, N and S elements must exist in the pyrolysis and rearrangement process of macromolecules, resulting in the generation of $NO_2$ and $SO_2$. The SSD asphalt has a significant inhibitory effect on small molecules, among which the main causes of odor are $H_2S$ and SO, which is the main reason for reducing the odor intensity of asphalt fume. The mass to charge ratios in Fig 20 are 128, 154, 166, 178 and 202, which are representative polycyclic aromatic hydrocarbons (PAHs). From left to right: naphthalene, acenaphthylene, fluorene, anthracene and pyrene are known as probable, suspected, or typical carcinogens. Based on the ion current data, the addition of SBS, MOS and deodorant has a certain inhibitory effect, but it is almost stable at approximately 80%. In addition to the above two kinds of molecules, there are other organic volatile

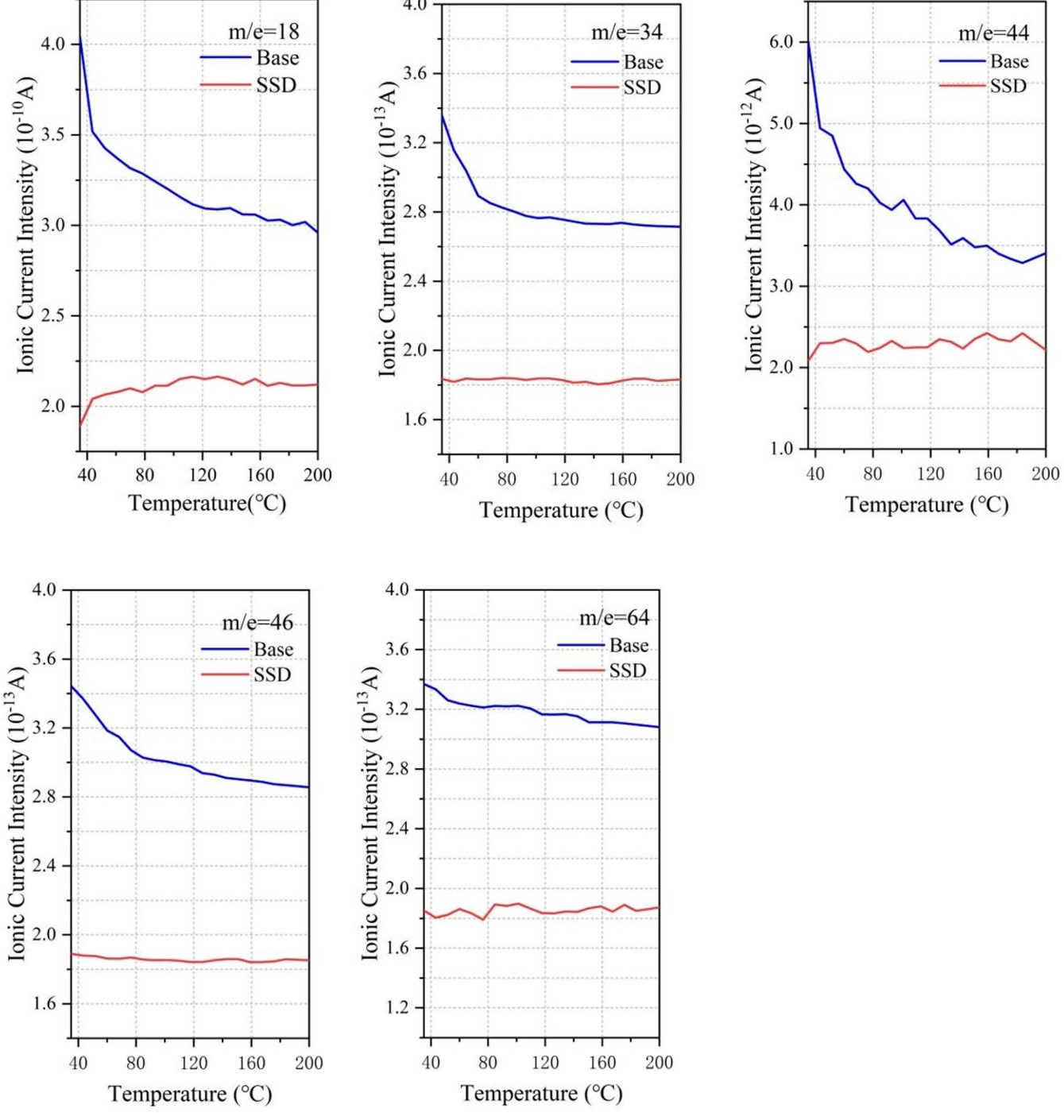

**Fig 19. Suppressive effect of SSD on small molecule volatiles.**

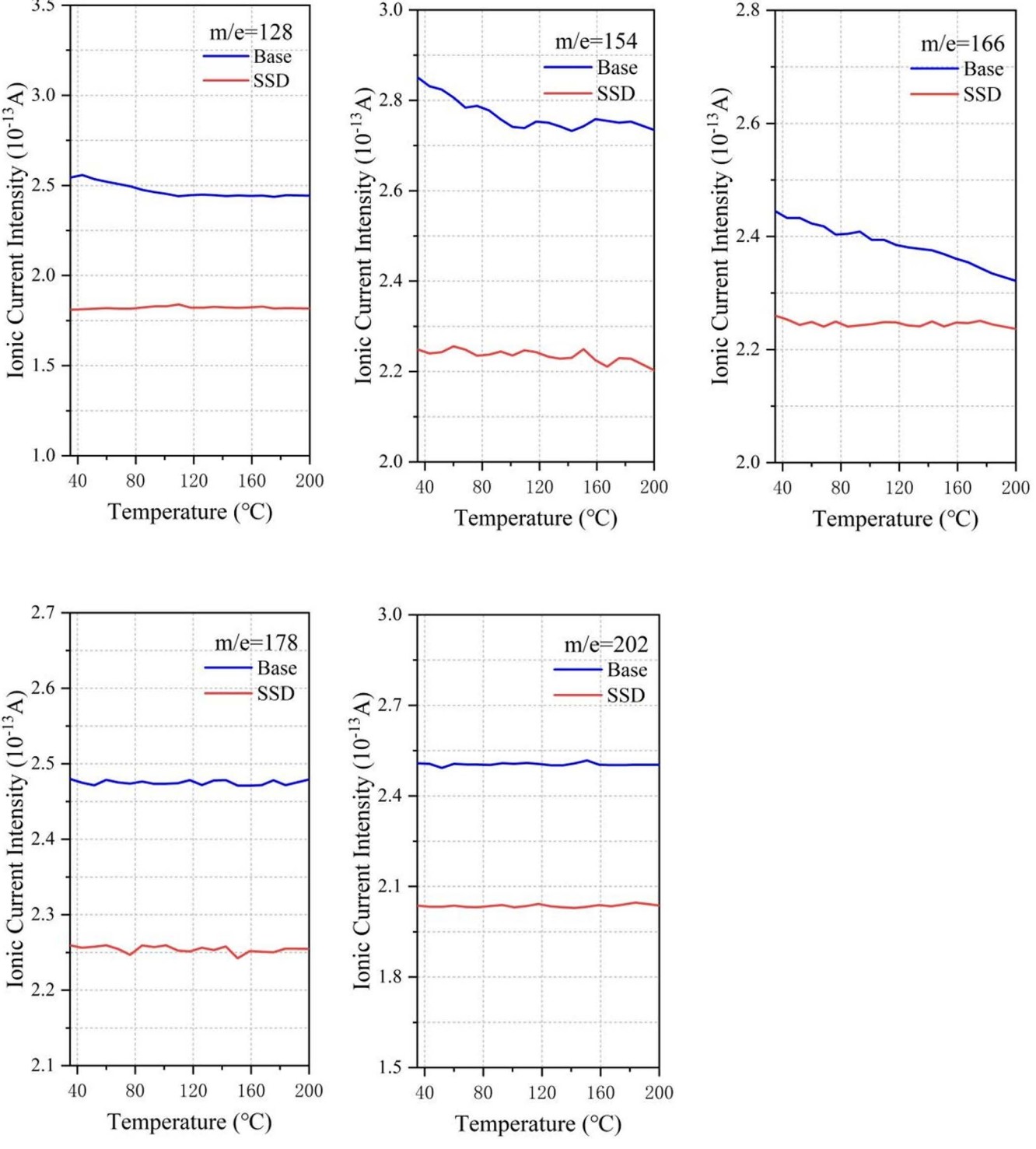

**Fig 20. Suppressive effect of SSD asphalt on PAHs.**

substances, including n-octane (m/e = 114), benzyl chloride (m/e = 126) and pentadecane (m/e = 212), as shown in Fig 21. Compared with the base asphalt, the addition of the three kinds of inhibitors in the SSD asphalt had a certain inhibitory effect on its amount, but the inhibitory effect was not as obvious as that of small molecular substances.

The spider diagram (Fig 22) was drawn based on the ratio of the sum of the ion current intensity diagram, so that the inhibition for various substances in asphalt fume can be seen more intuitively.

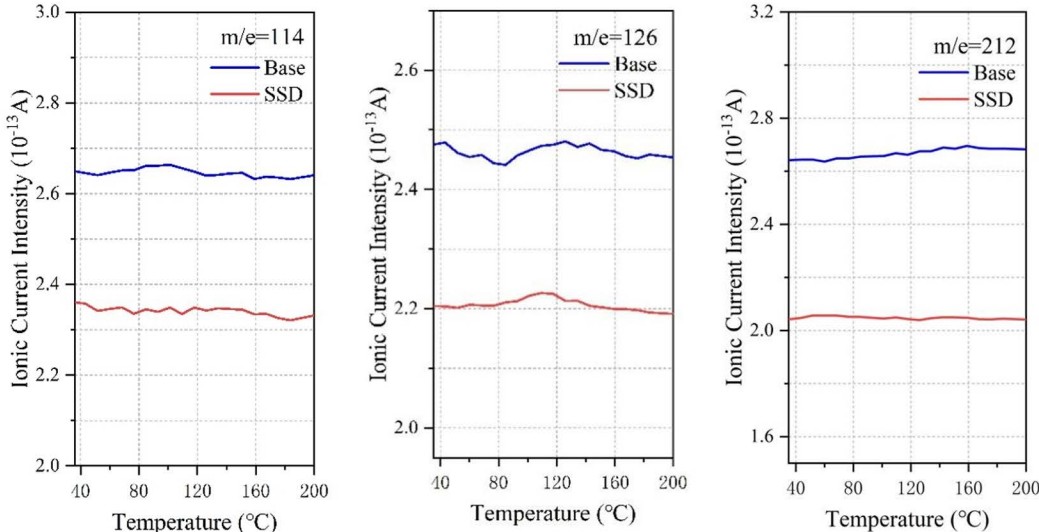

**Fig 21. Suppressive effect of SSD asphalt on Other organic volatiles.**

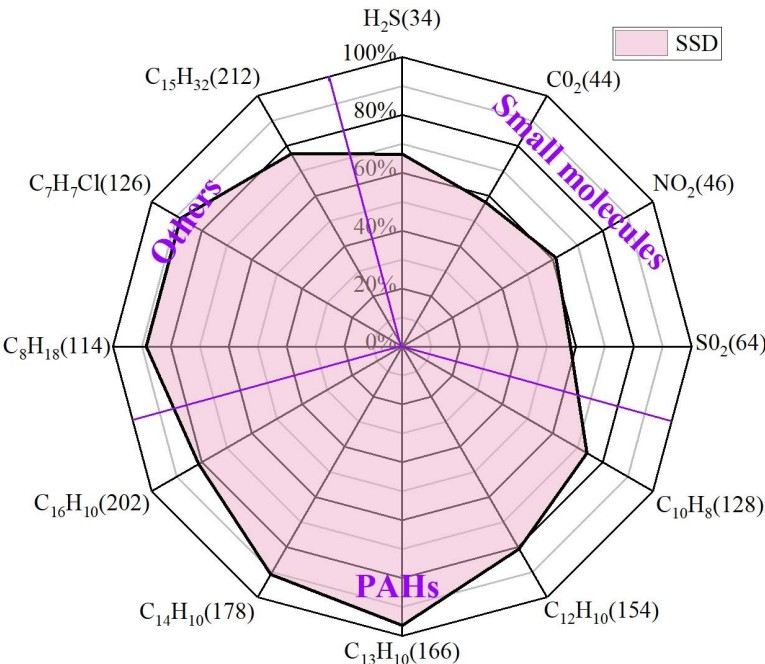

**Fig 22. Suppressive effect of SSD asphalt.**

## 4 Conclusion

In this paper, the modification of asphalt by SBS, MOS and deodorant was studied, on the premise of ensuring road performance of the asphalt, to solve the problem of fume emission of asphalt at high temperatures and eliminate the irritant odor of asphalt. According to the above results and discussions, the following conclusions can be drawn.

1. The release characteristics of asphalt fume were studied. Results show that heating temperature has the greatest influence on the asphalt fume emission. The asphalt fume emission and its growth rate increase significantly with the increase of temperature. The heating time is positively correlated with fume emission, but the increasing rate decreases gradually. With the increase of mixing speed, the release amount of asphalt fume increases, and the growth rate increases initially and then decreases.

2. SBS and MOS whiskers can effectively reduce the release amount of asphalt fume, and have synergistic benefits. By shear modification, 0.15‰ deodorant, 3%SBS and 3% MOS whisker are dispersed in the base asphalt, and a new type of SSD asphalt is obtained, which can effectively degrade 53% of asphalt fume and has no irritant odor at high temperatures.

3. According to the TG-DSC analysis, SSD asphalt exhibits higher thermal stability and lower temperature sensitivity compared to the base asphalt. At high temperatures, SSD asphalt is more stable, thus reducing the amount of the asphalt fume released. Besides that, SSD asphalt mixture also has good road performance.

4. The inhibition of SSD asphalt for small molecules, polycyclic aromatic hydrocarbons and other organic volatiles was characterized by ionic current intensity. The results show that the small molecules in the SSD asphalt had an excellent inhibitory effect, while its inhibitory effect on polycyclic aromatic hydrocarbons and other organic volatile substances was maintained at approximately 80%.

## Supporting information

**S1 File.** S1 Relevant data underlying Fig 5. S2 Relevant data underlying Fig 6. S3 Relevant data underlying Fig 7. S4 Relevant data underlying Fig 8. S5 Relevant data underlying Fig 11. S6~8 Relevant data underlying Fig 12. S9~11 Relevant data underlying Fig 13. S12 Relevant data underlying Fig 16. S13 Relevant data underlying Fig 21.
(ZIP)

## Author contributions

**Conceptualization:** Guang Yang.

**Data curation:** Guang Yang.

**Formal analysis:** Guang Yang.

**Funding acquisition:** Guang Yang.

**Investigation:** Guang Yang.

**Methodology:** Yongli Xu.

**Resources:** Xiaolei Jiao.

**Software:** Xiaolei Jiao.

**Supervision:** Xiaolei Jiao.

**Validation:** Yiming Li.

**Visualization:** Yongli Xu, Yiming Li.

**Writing – original draft:** Guang Yang.

**Writing – review & editing:** Yongli Xu.

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
