## [Decision Letter · Decision Letter 0]

PONE-D-25-03422Study on the Releasing Regularity of Asphalt Fume and Its Suppression TechnologyPLOS ONE

Dear Dr. Xu,

Thank you for submitting your manuscript to PLOS ONE. After careful consideration, we feel that it has merit but does not fully meet PLOS ONE’s publication criteria as it currently stands. Therefore, we invite you to submit a revised version of the manuscript that addresses the points raised during the review process. 

We look forward to receiving your revised manuscript.

Kind regards,

Mayank Sukhija

Academic Editor

PLOS ONE

Journal Requirements:

The authors would like to express their sincere gratitude to the Doctoral Innovation Project of Manuscript Click here to access/download;Manuscript;Manuscript.docx Northeast Forestry University (2572022AW55) for financial support

the Doctoral Innovation Project of Northeast Forestry University (2572022AW55) received by Guang Yang

The funders had no role in study design, data collection and analysis, decision to publish, or preparation of the manuscript

5. Please ensure that you refer to Figure 2 and 17 in your text as, if accepted, production will need this reference to link the reader to the figure.

Additional Editor Comments :

Kindly incorporate the comments suggested by the Reviewers. They are recommending aggressive revision before further consideration.

Reviewers' comments:

Reviewer's Responses to Questions

**Comments to the Author**

1. Is the manuscript technically sound, and do the data support the conclusions?

Reviewer #1: Partly

Reviewer #2: Yes

2. Has the statistical analysis been performed appropriately and rigorously? 

Reviewer #1: No

Reviewer #2: Yes

3. Have the authors made all data underlying the findings in their manuscript fully available?

Reviewer #1: No

Reviewer #2: Yes

4. Is the manuscript presented in an intelligible fashion and written in standard English?

Reviewer #1: Yes

Reviewer #2: Yes

5. Review Comments to the Author

Reviewer #1: This manuscript is a study of the release patterns and inhibition techniques for asphalt fumes. Smoke-suppressing and deodorizing asphalt was prepared by adding SBS, MOS and deodorizer to the matrix asphalt. The smoke inhibition and deodorization of matrix asphalt by smoke inhibitor and deodorizer were also analyzed by mass method, odor intensity grading method, FTIR, SEM and TG-MS. However, the manuscript has more detailed problems, and the specific modifications are as follows:

1. There are some problems with the structure and logic of the introduction section. The introduction of the fume collection device seems rather abrupt and does not form an effective connection with the contextual content. In addition, several concluding statements lacked the support of corresponding literature citations.

2. The manuscript is inaccurate in its description of Table I. Is SBS a modifier or a bitumen? What is the purpose of mentioning different matrix bitumens in Table I? Were these asphalts set up as controls? The tensile speed for the ductility test should be 50 mm/min. In addition, the specifications for the modifier MOS and deodorizer need to be listed.

3. Is the use of PTFE filter membranes supported by the results of other studies?

4. The description of the test program in Table 2 is very vague, please redraw the table.

5. In “2.2methods for asphalt fume”, please add the description of TG test and DSC test methods.

6. In subsection 3.1, the author's description of Figures 6 and 7 is wrong. The release of asphalt fume tends to increase with the increase of heating time and heating temperature.

7. In the summary of the release pattern of asphalt smoke, the release of asphalt smoke is greater when the temperature is 180°C and the time is 5 h. Why were the test conditions finally determined at a heating temperature of 160°C and a time of 3 h?

8. In Fig. 12, it is suggested to add obvious labeling to point out the location where the MOS whiskers are exposed.

9. In both the FTIR and TG tests, the tests were carried out on five types of bitumen. Therefore, it is suggested that the SEM test, DSC test and TG-MS test should also be carried out on five kinds of bitumen to ensure the comprehensiveness of the test.

10. The road performance of modified asphalt is mentioned several times in the manuscript, but the results of the effect of various modifiers on the road performance of asphalt are not reflected in the whole text, so it is suggested that the authors add.

11. “SEM” is spelled incorrectly several times in the manuscript, and it is recommended that the authors check the whole text.

12. In subsection 3.7, it is mentioned that SSD asphalt has low temperature sensitivity, while in conclusion 4, it is mentioned that SSD has excellent temperature sensitivity, which is contradictory.

13. The evaluation of the deodorizing effect of SSD asphalt in this manuscript relies only on the Odor Intensity Rating Method (OIRM), but the results of this test are not listed in the text and need to be supplemented. However, the results of this test are not listed in the text, which should be added. Moreover, the odor intensity rating method mainly comes from the personal feeling of each volunteer, which is subjective. The TG-MS test can qualitatively and quantitatively analyze the substances in asphalt fumes, and it is suggested that the authors add more test data to evaluate the deodorizing effect of deodorizers.

14. The result from Fig. 9 is that the smoke inhibition rate reaches 53% when the MOS doping is 3.0% and SBS doping is 4.0%, which is not in line with the description of the authors.

15. The dosage of deodorant was determined without specific experimental basis, and it is recommended that the authors add the effect of different dosages on asphalt performance to arrive at the optimum dosage of deodorant.

16. The writing of the conclusion is too long, it is recommended that the author streamline the conclusion and refine the core points.

Reviewer #2: This work contributes a valuable study on the performance of styrene-butadiene-styrene (SBS), magnesium oxysulfate (MOS) whisker, and deodorant in inhibition of the asphalt emissions, which enables better analysis of fume suppression effect of additives in asphalt. And multiple technologies were used in this study, including TG, DSC, TG-MS, SEM, and FTIR. According to the current manuscript, there are some points the authors need to consider for improving the manuscript, some detailed comments are shown as below:

1. It seems like the aerosol portion of asphalt fumes is neglected in the proposed asphalt fume collection device, yet it takes up a major portion. Why didn’t the authors consider the aerosol portion in this step?

2. What are the differences between the asphalt fume collection devices of the authors and others?

3. According to Figure 6, the mass of asphalt fumes decreased with stirring time. Did the authors consider the effect of modification process and pre-heating process on the inhibition of asphalt fumes?

4. The authors developed an asphalt fume collection device, how did the authors ensure the effect of this device?

5. How did the authors control the air flow rate?

6. PLOS authors have the option to publish the peer review history of their article (what does this mean? ). If published, this will include your full peer review and any attached files.

**Do you want your identity to be public for this peer review?** For information about this choice, including consent withdrawal, please see our Privacy Policy .

Reviewer #1: No

Reviewer #2: No

---

## [Author Response · Author response to Decision Letter 1]

14 Apr 2025

he point-by-point responses to the kind reviwers and the nice editor are included in the attached file entitled response to the reviewers.

---

## [Decision Letter · Decision Letter 1]

PONE-D-25-03422R1Study on the Releasing Regularity of Asphalt Fume and Its Suppression TechnologyPLOS ONE

Dear Dr. Xu,

Thank you for submitting your manuscript to PLOS ONE. After careful consideration, we feel that it has merit but does not fully meet PLOS ONE’s publication criteria as it currently stands. Therefore, we invite you to submit a revised version of the manuscript that addresses the points raised during the review process.

We look forward to receiving your revised manuscript.

Kind regards,

Mayank Sukhija

Academic Editor

PLOS ONE

Journal Requirements:

Additional Editor Comments:

Please revise the manuscript as per the reviewers comments. Kindly imrpove the quality of the manuscript in terms of typos and grammar.

Reviewers' comments:

Reviewer's Responses to Questions

**Comments to the Author**

1. If the authors have adequately addressed your comments raised in a previous round of review and you feel that this manuscript is now acceptable for publication, you may indicate that here to bypass the “Comments to the Author” section, enter your conflict of interest statement in the “Confidential to Editor” section, and submit your "Accept" recommendation.

Reviewer #2: All comments have been addressed

Reviewer #3: All comments have been addressed

Reviewer #4: All comments have been addressed

2. Is the manuscript technically sound, and do the data support the conclusions?

Reviewer #2: Yes

Reviewer #3: Yes

Reviewer #4: Yes

3. Has the statistical analysis been performed appropriately and rigorously? 

Reviewer #2: Yes

Reviewer #3: Yes

Reviewer #4: No

4. Have the authors made all data underlying the findings in their manuscript fully available?

Reviewer #2: Yes

Reviewer #3: Yes

Reviewer #4: Yes

5. Is the manuscript presented in an intelligible fashion and written in standard English?

Reviewer #2: Yes

Reviewer #3: Yes

Reviewer #4: No

6. Review Comments to the Author

Reviewer #2: The manuscript is revised according to my comments, and the revision is satisfactory and suitable for the publication.

Reviewer #3: In this manuscript, the release characteristics of asphalt fumes and their suppression techniques were analyzed. Modified asphalt with smoke suppression and deodorization functions were prepared by adding SBS, MOS and deodorant to the base asphalt. In addition, the asphalt under the action of smoke suppressants and deodorants was systematically studied by various methods, and its performance in smoke suppression and deodorization was discussed in detail. Generally, there are no major issues with the manuscript, but it can be improved in the following aspects.

1. The sources are considered old in terms of history and not the scientific content. It is better to add some modern sources.

2. Please increase the resolution of the images installed in the search.

3. The conclusion is a little long and it fails to highlight the actual contribution of this study completely. It is recommended to rewrite it.

Reviewer #4: The authors have answered the reviewers' comments. The structure and methodology of the paper are both reasonable. However, some revisions are suggested based on the following comments.

1. The English grammar needs to be carefully checked and improved throughout the manuscript, e.g. by a professional English language editor. The current version is difficult for readers to comprehend.

2. In the test section, the flowchart may be included to clearly demonstrate the research scheme of the paper.

3. It is recommended to use the same range for the vertical axis of all figures in Figure 18 ~ Figure 20.

4. To guarantee the accuracy of smoke testing, does the base asphalt utilized in the preparation of modified asphalt share the same heating history?

5. Kindly ensure that the paper adheres to the appropriate format and line - spacing requirements. Have the line numbers been accurately assigned?

7. PLOS authors have the option to publish the peer review history of their article (what does this mean? ). If published, this will include your full peer review and any attached files.

**Do you want your identity to be public for this peer review?** For information about this choice, including consent withdrawal, please see our Privacy Policy .

Reviewer #2: No

Reviewer #3: No

Reviewer #4: No

---

## [Author Response · Author response to Decision Letter 2]

27 Jun 2025

Please refer to the attachment 'Response to reviewers'

---

## [Editor Report · Decision Letter 2]

Study on the Releasing Regularity of Asphalt Fume and Its Suppression Technology

PONE-D-25-03422R2

Dear Dr. Xu,

We’re pleased to inform you that your manuscript has been judged scientifically suitable for publication and will be formally accepted for publication once it meets all outstanding technical requirements.

Kind regards,

Mayank Sukhija

Academic Editor

PLOS ONE
---

## [Editor Report · Acceptance letter]

PONE-D-25-03422R2

PLOS ONE

Dear Dr. Xu,

I'm pleased to inform you that your manuscript has been deemed suitable for publication in PLOS ONE. Congratulations! Your manuscript is now being handed over to our production team.

Kind regards,

on behalf of

Dr. Mayank Sukhija

Academic Editor

PLOS ONE